# Probabilistic Forecasting of Radiation Exposure for Spaceflight

## Abstract

Extended human presence beyond low-Earth orbit (BLEO) during missions to the Moon and Mars will pose significant challenges in the near future. A primary health risk associated with these missions is radiation exposure, primarily from galatic cosmic rays (GCRs) and solar proton events (SPEs). While GCRs present a more consistent, albeit modulated threat, SPEs are harder to predict and can deliver acute doses over short periods. Currently NASA utilizes analytical tools for monitoring the space radiation environment in order to make decisions of immediate action to shelter astronauts. However this reactive approach could be significantly enhanced by predictive models that can forecast radiation exposure in advance, ideally hours ahead of major events, while providing estimates of prediction uncertainty to improve decision-making. In this work we present a machine learning approach for forecasting radiation exposure in BLEO using multimodal time-series data including direct solar imagery from Solar Dynamics Observatory, X-ray flux measurements from GOES missions, and radiation dose measurements from the BioSentinel satellite that was launched as part of Artemis 1 mission. To our knowledge, this is the first time full-disk solar imagery has been used to forecast radiation exposure. We demonstrate that our model can predict the onset of increased radiation due to an SPE event, as well as the radiation decay profile after an event has occurred.

## 1   Introduction

Human spaceflight is projected to expand significantly in the coming years [2]. Unlike the majority of previous missions, upcoming crewed launches will venture beyond low-Earth orbit (BLEO). A notable example is the Artemis program, which aims to establish a permanent lunar base as part of its long-term objectives [5]. This program marks a significant milestone in human space exploration and is intended to serve as a precursor to deeper space expeditions, including missions to Mars.

A critical challenge that must be addressed when sending astronauts into BLEO is the mitigation of space radiation exposure. Space radiation poses serious risks to astronaut health, particularly during extravehicular activities (EVAs) [1]. Prolonged exposure can result in DNA damage, increasing the likelihood of cancer and degenerative diseases later in life. In the short term, astronauts may face acute radiation syndrome, a potentially life-threatening condition [17]. As such, space radiation is considered one of the greatest hazards for astronauts in BLEO [4].

Space radiation primarily originates from two sources: galactic cosmic rays (GCRs) and solar energetic particles (SEPs). GCRs are high-energy particles that originate outside the Solar System and consist of protons, electrons, alpha particles, and heavier ions. SEPs, on the other hand, are high-energy particles produced by solar activity, such as solar flares and coronal mass ejections (CMEs), and are primarily composed of protons. Both GCRs and SEPs are influenced by the solar cycle. During solar maxima, the frequency of SEP events increases, while the intensity of GCRs decreases. Overall, GCR intensity is generally anti-correlated with the solar cycle [14].

Submitted to 38th Conference on Neural Information Processing Systems (NeurIPS 2024). Do not distribute.

NASA has developed a variety of tools to monitor space radiation and estimate the biological impacts of radiation events. More recently these include the *SEP Scoreboard*,[1] a unified portal running an ensemble of SEP models developed by the space radiation community at various institutions [15], and the *Acute Radiation Risks Tool* (ARRT), which is a model that estimates the biological impacts of space radiation on astronauts during an SEP event [9]. These tools are used in decision-making for immediate action such as canceling EVAs or moving astronauts to shielded areas once an increase in radiation levels is detected. However, at the moment, these tools are not integrated, and ARRT only makes radiation predictions after notification of an actual SEP event. Additionally, the majority of radiation data used to develop ARRT comes from the International Space Station (ISS), which operates in low Earth orbit (LEO) [16]. Objects in LEO experience less severe radiation on average than objects in BLEO because of shielding from Earth's magnetic field [3].

Advancing the state-of-the-art through predictive models that can forecast radiation exposure ahead of major SEP events, ideally hours in advance, would complement existing tools and significantly enhance the decision-making processes to ensure safety of astronauts in BLEO. This area is ripe for machine learning (ML) applications given large datasets of solar data, including petabyte-level high-resolution image data from the Solar Dynamics Observatory (SDO), and radiation data made available through the NASA Open Science for Life in Space program's RadLab data portal [8].

In this paper we propose an ML approach for forecasting space radiation, leveraging time-series data from BioSentinel [13], a NASA mission launched as part of Artemis 1 that measures the radiation environment in an Earth-trailing heliocentric orbit, X-ray measurements from the Geostationary Operational Environmental Satellite (GOES) [12] missions, and crucially SDO images observing the Sun across several extreme ultraviolet wavelengths [11]. We demonstrate how our model can be used to predict both the onset of increased radiation hours ahead of time, as well as the radiation decay profile after an event has occurred. Our work represents a proof-of-concept that we believe can be extended to forecast radiation levels at different locations in the Solar System and be used to complement existing radiation monitoring tools during human space missions.

## 2   Data

**Solar observations**   SDO [11] is a state-of-the-art spacecraft designed to capture high-resolution images of the Sun that document solar phenomena responsible for SEPs, such as CMEs and solar flares. Since its launch in 2010, SDO has produced over 12 petabytes of raw data. Recent efforts such as SDOML [7] have leveraged this vast data source to make it more accessible for ML applications. In this work, we have curated a novel ML-ready dataset, called *SDOML-lite*, which reduces the cadence to 15 minutes, focuses on a subset of Atmospheric Imaging Assembly (AIA) wavelengths (131, 171, 193, 211, and 1600 Å), and includes only one Helioseismic and Magnetic Imager (HMI) channel— specifically, the line-of-sight magnetogram. This data reduction facilitates a more manageable dataset while retaining key information related to solar activity. Additionally, we incorporate solar X-ray flux data from the GOES satellites [12], as X-rays are critical precursors to SEP events.

**Radiation observations**   Measurements of absorbed dose[2] rate of radiation have been made by multiple space missions of different durations at a handful of locations in LEO and BLEO. For our initial work, we make use of the radiation measurements from the currently ongoing BioSentinel mission [13] deployed in heliocentric orbit at 1 AU trailing behind Earth. Data from BioSentinel are available through the NASA RadLab portal [8], which includes various measurements from the ISS and several spacecraft at BLEO locations. We down-sampled the BioSentinel 1-minute cadence to 15 minutes to match the SDOML-lite cadence.

**Generating aligned time series**   The dataset for our task is generated by sliding a time window of 20 hours (a sequence of $20 \times 4 = 80$ timestamps given the 15-minute cadence) over the period between 16 Nov 2022 and 14 May 2024, based on the snapshot of BioSentinel data that we have access to. The window is divided into two parts: the context window of length $c$, containing the data used as input, and the prediction window of length $p$, containing the data used as the prediction target. For time $t$, the context window covers time steps $(t - c) \ldots t$ and includes SDO images, BioSentinel radiation data and GOES X-ray fluxes, while the prediction window covers time steps $(t + 1) \ldots (t + p)$ and includes BioSentinel radiation data and GOES X-ray (Figure 1). In our

---

[1] https://ccmc.gsfc.nasa.gov/scoreboards/sep/

[2] Energy deposited by ionizing radiation per unit of mass. Its SI unit is the gray (Gy) equal to 1 Joule of energy absorbed per kilogram of matter.

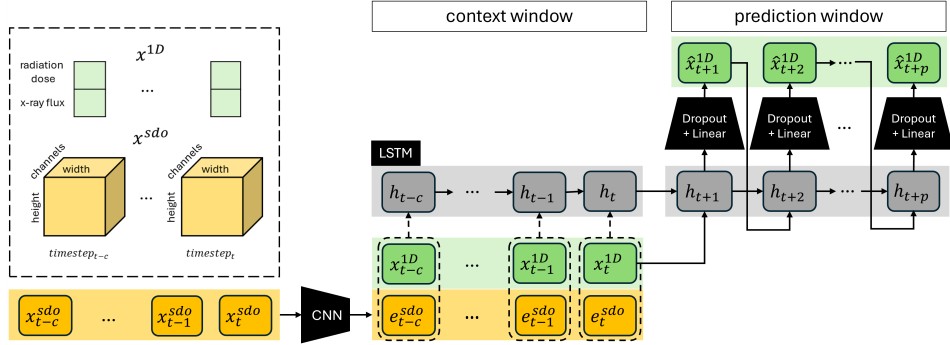

Figure 1: The overall model architecture and the construction of the context and prediction windows. $x^{sdo}$ and $e^{sdo}$ denote the SDO images and their embeddings through a CNN architecture; $x^{1D}$ denote the vectors of univariate time series data, including radiation dose rate and X-ray flux.

experiments we take $c = p = 10$ hours. If a timestamp is missing in one of the windows for either the solar observations or the radiation measurements, then that data point is discarded. Datapoints within certain intervals of time are kept as a hold-out set for validation and testing. These hold-out intervals are chosen using the NOAA Solar Energetic Particle (SEP) events catalog, which maintains the historical record of SEP events. Example data are visualized in Figures 3, 4, and 5 in the appendix.

## 3  Method

The model, shown in Figure 1, is composed of three main components: (1) a convolutional neural network (CNN) that is used to encode SDO images $x^{sdo} \in \mathbb{R}^{6 \times 512 \times 512}$ into a lower-dimensional space $e^{sdo} \in \mathbb{R}^{1024}$; (2) a long short-term memory (LSTM) network for encoding the context window of $e^{sdo}$ concatenated with univariate time series $x^{1D}$; and (3) another LSTM used to forecast the radiation dose rate in the prediction window, which is initialized with the final hidden state of the context LSTM representing an overall embedding of the whole context window. The LSTM for the prediction window is constructed such that its input and output at each time step are $x^{1D}$, the univariate time series of radiation and X-ray data at each time step, i.e., it does not deal with SDO data. This setup allows the prediction LSTM to be used in an autoregressive manner, where the output at each time step at prediction time can be used as input for the next time step allowing us to extend the predictions at test time to arbitrary numbers of future time steps. The model is trained to minimize the mean squared error between the predicted and actual radiation and X-ray values in the prediction window. For the experiments in this paper, we used a CNN configuration with three convolutional blocks with kernel size 3 and 64, 128, 128 channels, using max-pooling, followed by two linear layers that bring the output size to 1024. The two LSTMs have hidden state size 1024 and a stack of 2 layers. The model overall has 116,671,170 trainable parameters and is trained using the Adam optimizer [10] with a learning rate of $10^{-4}$ and a batch size of 4, using an Nvidia A100 80GB GPU. The low batch size was necessitated by the large memory requirements due to working with long sequences of 6-channel SDO images.

We use Monte Carlo (MC) dropout approximation [6] to produce predictive distributions for the univariate radiation and X-ray data in the prediction window. MC dropout corresponds to sampling from the posterior distribution of the model weights given the data used during training, which allows us to estimate the model uncertainty at test time. At test time, given data in the context window $(t - c) \ldots t$, we sample the model multiple times with different dropout masks to obtain an empirical predictive distribution for the radiation dose rate and X-ray fluxes at each time step in the prediction window $(t + 1) \ldots (t + p)$. Note that due to the autoregressive nature of the prediction LSTM, the prediction window can be of arbitrary size $p$ during inference time. This allows us to produce extended forecasts that last up to several days.

## 4  Results

In Figure 2 we show the forecasting results for one of the hold-out intervals in the test set, corresponding to an SEP event that took place in May 2024. Based on the MC dropout samples from the

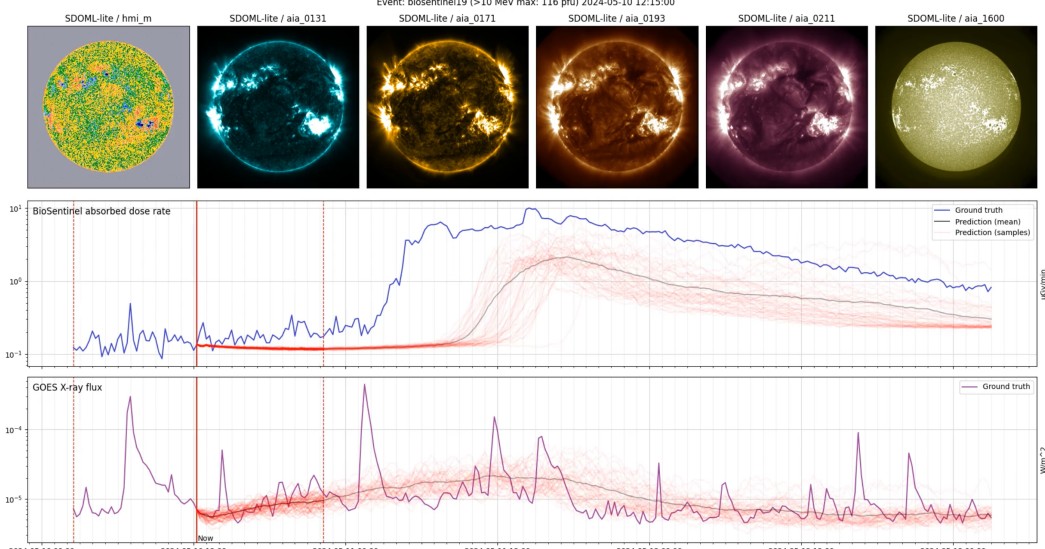

Figure 2: Model predictions (red with their mean in black) versus the ground truth (blue) in a hold-out interval around an SEP event occurring in May 2024 from the test set. The solid vertical red line indicates the "now" time after which the prediction is performed. The dashed vertical red line to the left of the "now" line indicates the beginning of the context window. The dashed vertical red line to the right of the "now" line indicates the end of the prediction window size used during training, but this window can be extended arbitrarily at test time due to the autoregressive nature of the model.

posterior predictive distribution, each red line shows one possible unfolding of the event during the prediction window. In this figure, instead of representing only the mean and variance of the predictive distribution, we choose to show the full set of samples to illustrate the model's uncertainty with better detail. Based on the information in the context window, the model can predict the existence of increased radiation levels hours ahead of time. While not being able to predict the exact onset time, this result is significant as it was made possible due to the inclusion of solar images in the model. This is a result we tested in an ablation study where we removed the solar images from the input and found that the model did not predict an onset at all. We also note that the model can predict the shape of the radiation profile approximately, but it underestimates the intensity of the radiation. Figure 6 in the appendix shows the model's performance in forecasting the radiation decay profile after the event's peak has occurred. The model does remarkably well in predicting the radiation decay profile, showing that it can be used as a forecasting tool to estimate when the environment will become safe again, which is a crucial piece of information for decision-making in this context.

## 5    Conclusions

**Contributions**    In this work, we have introduced a novel ML-based method to forecast radiation levels for human spaceflight. Our model leverages solar observations (images of the photosphere and corona, and X-ray fluxes) and past measurements of absorbed dose rates. This same procedure is then repeated iteratively. The method also offers a way to estimate the model uncertainties using the probabilistic interpretation of Dropout. We showed that the model can be used as an early warning system to predict future increases in radiation levels and, once high radiation levels are measured, it can be used to forecast when the environment will become safe again. To our knowledge, this is the first model that uses solar images to directly forecast radiation dose.

**Further work**    We are currently working on extensions of this model that incorporates radiation data from other instruments including CRaTER (Cosmic Ray Telescope for the Effects of Radiation) onboard the Lunar Reconnaissance Orbiter (LRO) and RAD (Radiation Assessment Detector) onboard the Curiosity rover on Mars. Our ultimate goal is to incorporate trained models into real-time forecasting systems that can be used to monitor radiation levels in the solar system and provide early warnings to astronauts during BLEO missions.

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

# 6 Appendix

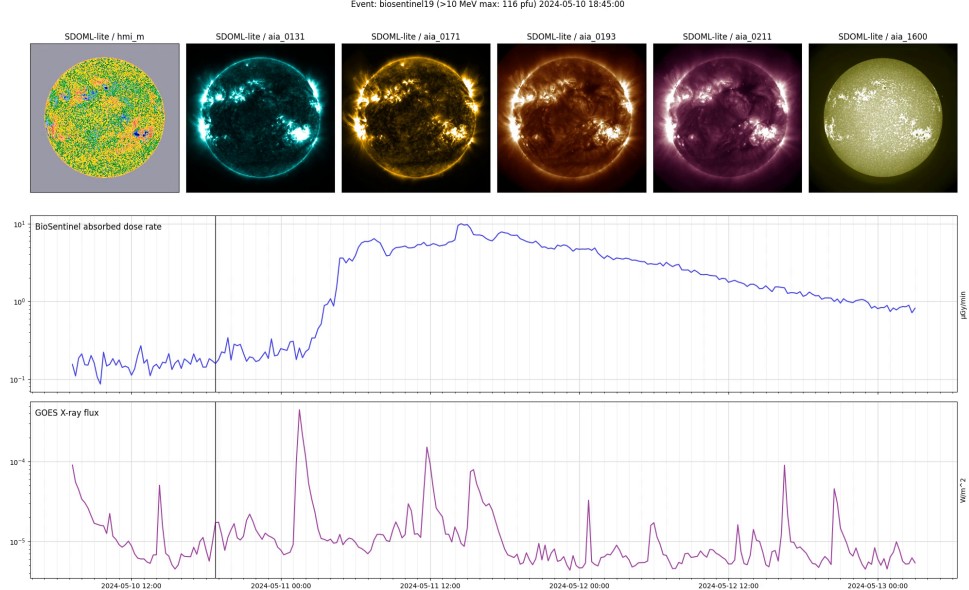

Figure 3: *Top row:* SDO imagery ($x_t^{sdo}$) at t = 2024-05-10 18:45 before the onset of the SEP event. *Middle & bottom rows:* Radiation dose and X-ray flux ($x^{1D}$) from t = 2024-05-10 07:00 to 2024-05-13 03:00.

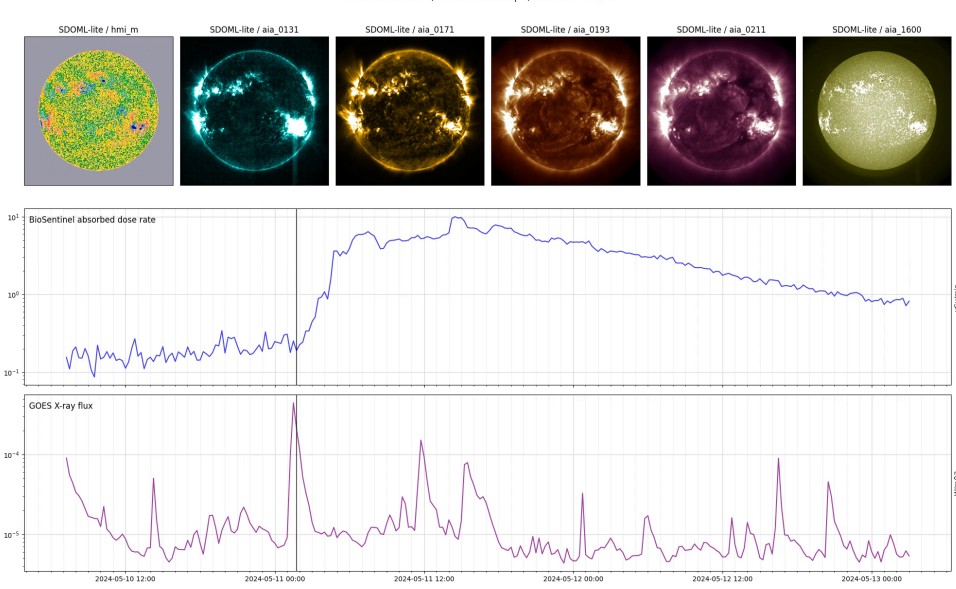

Figure 4: *Top row:* SDO imagery ($x_t^{sdo}$) at t = 2024-05-11 01:45. *Middle & bottom rows:* Radiation dose and X-ray flux ($x^{1D}$) from t = 2024-05-10 07:00 to 2024-05-13 03:00. Notice the flare occurring at the south-east region of the solar disk prominently visible in the aia_0131 channel *(top row, second column)* coinciding with a prominent peak in the X-ray flux right before the radiation peak onset.

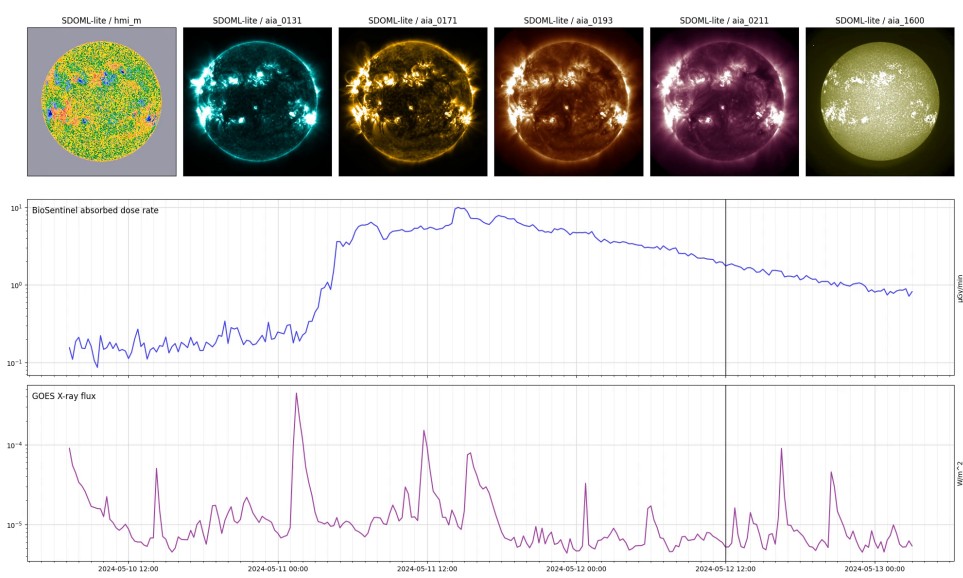

Figure 5: *Top row:* SDO imagery ($x_t^{sdo}$) at t = 2024-05-12 12:00 as the radiation decays. *Middle & bottom rows:* Radiation dose and X-ray flux ($x^{1D}$) from t = 2024-05-10 07:00 to 2024-05-13 03:00.

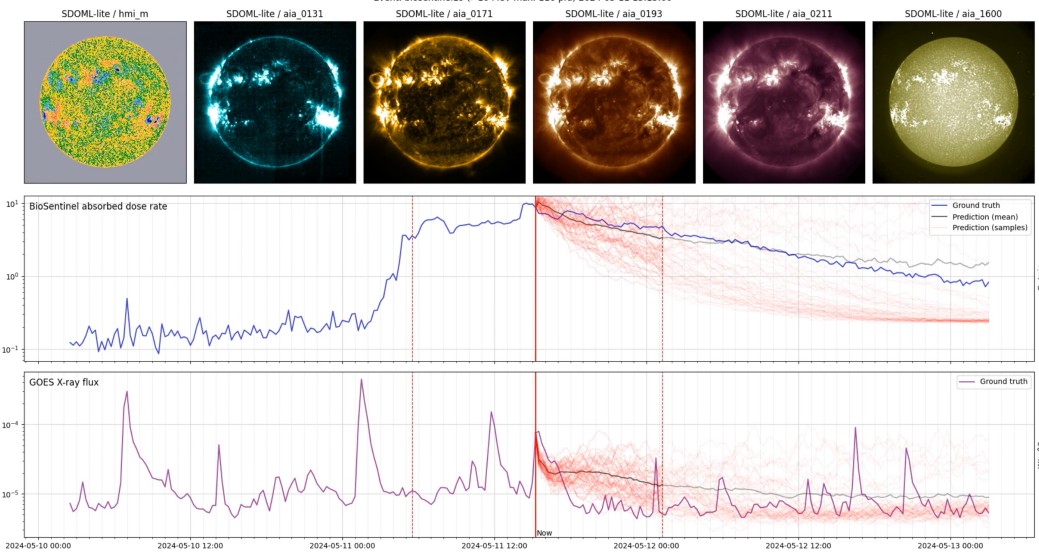

Figure 6: Post-event prediction of the radiation decay profile for the event shown in Figure 2. Model predictions (red with their mean in black) versus the ground truth (blue) in a hold-out interval around an SEP event occurring in May 2024 from the test set. The solid vertical red line indicates the "now" time after which the prediction is performed. The dashed vertical red line to the left of the "now" line indicates the beginning of the context window. The dashed vertical red line to the right of the "now" line indicates the end of the prediction window size used during training, but this window can be extended arbitrarily at test time due to the autoregressive nature of the model.

