# OpenReview forum: "Probabilistic Forecasting of Radiation Exposure for Spaceflight"
_NeurIPS.cc/2024/Workshop/BDU — Submitted to NeurIPS BDU Workshop 2024_

### Official Review · Reviewer_RK76 · 2024-09-26
**Interesting application that lacks quantitative assessment**

**Rating:** 4
**Confidence:** 3

**Review:**

Clarity:
* Lines 70-71: what is the native cadence of the SDOML from which you reduce?
* Lines 73-74: how much more “manageable” is the dataset? What's the gain?
* Line 81: which method do you use for downsampling?
* How many epochs do you train for?
* Lines 134-135: would have been useful to include the results of that ablation study in the Appendix.

Quality:
* “Generating aligned time series”  Did you experiment with different values for c and p? How often do the SEP events occur? Is there a class imbalance in the dataset, and a distribution shift between training and hold-out test set, since the latter focuses only on SEP events?
* From the auto-regressive nature of the model, there’s a risk of error accumulation over time, especially for long prediction horizons, it might be useful to discuss mitigation strategies for this issue. Although from Figure 2 it looks like the uncertainties are not higher the further you go from the context window, might the model be overconfident?
* From Figure 2, it looks like the model struggles to model peaks, wouldn’t that be an issue?
* There is a clear lack of reported metrics. What is the train/val/test MSE? Is there a simpler model you can compare against? For assessment, we only have access to one qualitative example, in Figure 2, which makes it hard to evaluate the relevance of the approach.


The paper is an interesting proof of concept (as the authors mention) and application of Bayesian method (Monte Carlo dropout) — however it lacks quantitative metrics on which to evaluate the quality of the work.

---

### Official Review · Reviewer_9Bi9 · 2024-10-06

**Rating:** 4
**Confidence:** 4

**Review:**

# Summary
This is an applied work that describes a model to predict the onset of solar proton events (SPEs). The work produces as part of its artifacts, a dataset collating data of the required wavelength at fixed time periods from the solar dynamics observatory (SDO). They use a sliding time window of 20 hours to predict events in the next 10 hours.

Overall Assessment: Reasonably good paper, but the model needs aggregate results and not one stand-alone event to be judged. The one event studied from 10th May 2024 may be an exception where the model performs well (or poorly). I could not find analysis of other events/dates in the paper.

# Strengths
- I liked the fact that instead of showing only the mean and variance, the authors decided to show the full set of monte carlo roll-outs.
- The problem statement is of immense practical importance to space exploration. Having said that, I am unable to gauge what has already been tried in literature, as that is missing. My brief study over relevant literature indicates that the problem has been addressed before, though not through this method. Happy to be corrected on this.

# Weaknesses
- only one model was tried out. What are the ablations? Why not other architectures? Why is this a good architecture?
- Why did you choose to only show the one figure from May 2024? is this a special case? Did you not study all events from 16 Nov 2022 and 14 May 2024? What are the overall results?
- The authors say, " The model does remarkably well in predicting the radiation decay profile" but my perusal of Figure 6 (bottom part) in the appendix indicates that the model is lagging on actual decay, i.e. the ground truth decay is faster than that predicted by the model. Is my understanding incorrect?

# Questions for authors:
- Perhaps I didn't understand Figure 2 correctly, but does the absorbed dose rate plot not indicate that the model is a lagging indicator? i.e. the model did not early predict the onset, but rather was 8-12 hours lagging behind actuals? I tried reading through a couple of times, but was not clear on the take-aways.



# Comments, Suggestions, Typos:
- Please consider changing the term "SPE event" due to redundancy. (line 18)
- Please consider using pdf images.

---

### Decision · Program_Chairs · 2024-10-09

**Decision:**

Reject

**Comment:**

Unfortunately reviewers think this work is not yet ready. I encourage the authors to look at feedback - especially involving comprehensiveness of presented results from Reviewer 9Bi9 - incorporate it, improve the work, and resubmit to a future workshop at ICML or other conference.